# Comparative Analysis of the Antimicrobial Activity of Essential Oils and Their Formulated Microemulsions against Foodborne Pathogens and Spoilage Bacteria

**DOI:** 10.3390/antibiotics11040447

**Published:** 2022-03-25

**Authors:** Raffaella Campana, Mattia Tiboni, Filippo Maggi, Loredana Cappellacci, Kevin Cianfaglione, Mohammad Reza Morshedloo, Emanuela Frangipani, Luca Casettari

**Affiliations:** 1Department of Biomolecular Sciences, University of Urbino, 61029 Urbino, Italy; mattia.tiboni@uniurb.it (M.T.); emanuela.frangipani@uniurb.it (E.F.); luca.casettari@uniurb.it (L.C.); 2School of Pharmacy, University of Camerino, 62032 Camerino, Italy; filippo.maggi@unicam.it (F.M.); loredana.cappellacci@unicam.it (L.C.); 3Université de Lorraine, CNRS, LIEC, F-57000 Metz, France; kevin.cianfaglione@gmail.com; 4Department of Horticultural Science, Faculty of Agriculture, University of Maragheh, Maragheh 55136-553, Iran; morshedlooreza@gmail.com

**Keywords:** essential oils, microemulsion, dynamic light scattering, antimicrobial efficacy, pathogens

## Abstract

The antimicrobial activity of several essential oils (EOs) and their related microemulsions (MEs) was investigated. EOs were obtained from *Cannabis sativa* L. cv CS (*C*. *sativa*), *Carum carvi* L. (*C. carvi*), *Crithmum* *maritimum* L. (*C. maritimum*), *Cuminum cyminum* L. (*C. cyminum*), x *Cupressocyparis leylandii* A.B. Jacks & Dallim. (*C. leylandii*), *Cupressus arizonica* Greene (*C. arizonica*), *Ferula assa-foetida* L. (*F. assa-foetida*)., *Ferula gummosa* Boiss. (*F. gummosa*), *Juniperus communis* L. (*J. communis*), *Juniperus* x *pfitzeriana* (Spath) P.A. Schmidt (*J. pfitzeriana*), *Pimpinella anisum* L (*P. anisum*). Preliminary screening revealed that *Cuminum cyminum*, *Crithmum maritimum*, and *Pimpinella anisum* (10% *v*/*v*) were effective against all tested microorganisms (*Escherichia coli* ATCC 35218, *Listeria monocytogenes* ATCC 7644, *Staphylococcus aureus* ATCC 29213, *Pseudomonas fluorescens* DSM 4358, and *Candida albicans* ATCC 10231), with growth inhibition diameter from 10 to 25 mm. These EOs were used to formulate the MEs with an average size < 50 nm and a good stability over 30 days. EOs’ antimicrobial activity was further enhanced in the MEs, with a generalized lowering of minimum inhibitory concentration (MIC) and minimum bactericidal concentration (MBC) values. *C. cyminum*-ME reached, in most cases, MIC two times lower (0.312%) than the corresponding EO (0.625%) and even eight times lower against *S. aureus* (0.156 vs. 1.25%). A more remarkable microbicide effect was noted for *C. cyminum*-ME, with MBC values eight times lower (from 0.312 to 0.625%) than the corresponding EO (from 2.5 to 5%). Overall, MEs resulted in an efficient system for EOs encapsulation, enhancing solubility and lowering concentration to exert antimicrobial efficacy.

## 1. Introduction

Essential oils (EOs) are naturally occurring mixtures of compounds, synthesized as secondary metabolites in different organs of aromatic plants. These molecules provide protection to the plants from external agents (e.g., UV light, herbivorous animals, insects, and pathogens) [1] and they are generally released after humidity variation or mechanical stimulation.

EOs are characterized by a high molecular reactivity and, therefore, they are accumulated in secretory glandules (specialized structures on plant surfaces) or in the internal cell vacuoles, to offer protection and prevent possible and dangerous interaction with producers’ vital parts [2]. The use of EOs as natural remedies in the treatment of infectious diseases dates back over 150 years. However, since nowadays there is a renewed interest in natural compounds, EOs have recently been reinvestigated as promising and sustainable resources for the development of new antimicrobials [3].

EOs are composed of volatile, lipophilic compounds, mostly represented by monoterpenes and sesquiterpenes, but also phenylpropanoids, aldehydes, and aliphatic hydrocarbons [1], with a wide spectrum of biological properties including antibacterial and antifungal activities [4,5,6,7,8,9]. About 10% of the 3000 known EOs are applied in different fields such as agronomy, cosmetics, healthcare, and pharmaceutics [10,11]. The biological activities of EOs are correlated to the complex interactions among the hundreds of compounds they contain. Nevertheless, external factors such as geographic and seasonal variations, environmental conditions, as well as the harvest period, drying processes, extraction technique, and storage can determine changes in their chemical composition [12]. In addition, volatility, poor water solubility, and instability in the presence of heat, light, and oxygen represent limiting factors for many EOs applications [13]. To overcome these limits, EOs encapsulation into different delivery systems including liposomes, cyclodextrins, solid lipid nanoparticles, micelles, and polymer-based nanocarriers could preserve and even enhance their activities. Among these formulation strategies, microemulsions (MEs) provide additional advantages such as an increased bioactivity due to the reduced subcellular size, and a better diffusion of the active compounds towards the target [14,15,16]. Indeed, the wetting ability of surfactants and the presence of emulsifiers can contribute to the antimicrobial and antibiofilm activities of EOs-based MEs [17,18,19].

EOs were obtained from *Cannabis sativa* L. cv CS (*C. sativa*), *Carum carvi* L. (*C. carvi*), *Crithmum maritimum* L. (*C. maritimum*), *Cuminum cyminum* L. (*C. cyminum*), x *Cupressocyparis leylandii* A.B. Jacks & Dallim. (*C. leylandii*), *Cupressus arizonica* Greene (*C. arizonica*), *Ferula assa-foetida* L. (*F. assa-foetida*), *Ferula gummosa* Boiss. (*F. gummosa*), *Juniperus communis* L. (*J. communis*), *Juniperus* x *pfitzeriana* (Spath) P.A. Schmidt (*J. pfitzeriana*), *Pimpinella anisum* L (*P. anisum*). The Apiaceae plants (e.g., *C. carvi*, *C. maritimum*, *C. cyminum*, *F. assa-foetida*, *F. gummosa* and *P. anisum*) are employed in different fields such as food and beverages, cosmetics and pharmaceutics [20]. In addition, they are considered as promising candidates for ecofriendly biocides for parasite and vector control [21].

The Cupressaceae plants, such as *C. leylandii, C. arizonica, J. Communis and J. pfitzeriana*, are used on an industrial level to make varnishes, solvents, paints, flavor and fragrances, food additive, perfumes, and pharmaceuticals [22]. Moreover, some Cupressaceae EOs were assessed for their insecticidal activity against three important insect species, such as *Spodoptera littoralis*, *Culex quinquefasciatus*, and *Musca domestica*, resulting in candidate ingredients in formulations of effective green insecticides [23]. *Cannabis sativa* (Cannabaceae) is gaining interest in different fields, such as organic agriculture, pharmaceuticals, cosmetics, and botanical insecticides [24]. Moreover, *C. sativa* EO displayed anti-inflammatory, antiprotozoal, and antioxidant effects [25].

The aim of the present work was to investigate and compare the antimicrobial activity of selected EOs and their ME formulations, against the representative foodborne pathogens (*E. coli* ATCC 35218, *L. monocytogenes* ATCC 7644, *S. aureus* ATCC 29213), as well as spoilage and contaminant microorganisms (*P. fluorescens* DSM 4358 and *C. albicans* ATCC 10231). The experimental design was subdivided into four distinct phases: (i) plants collection, extraction, and chemical characterization of eleven different EOs; (ii) selection of the most active EOs by agar well diffusion method (AWDM); (iii) EOs-microemulsion formulation (EO-ME) and characterization; (iv) comparison of the antimicrobial activity of EOs and related EOs-ME by MIC/MBC determination.

## 2. Results

### 2.1. Screening of EOs Antimicrobial Activity

The antimicrobial activity of each EO (10% *v*/*v*) was initially determined by AWDM (Table 1 and Figure 1). To select the most active ones, the presence of a growth inhibition diameter (GID) > 10 mm against all the examined microorganisms was considered as an index of antimicrobial activity.

As shown, among the examined EOs, *C. cyminum* was the most active against all tested bacteria, with GID ranging between 11 ± 1.0 mm for *E. coli* ATCC 35218 and 15 ± 1.4 mm for *S. aureus* ATCC 29213. Moreover, C. cyminum EO was able to highly inhibit the growth of *C. albicans* ATCC 10231 showing a GID value of 25 ± 1.2 mm, similar to that of fluconazole (50 µg). Another interesting activity was observed for the *C. maritimum* EO, with the highest GID values of 13 ± 0.4 mm and 25 ± 1.5 mm against *S. aureus* ATCC 29213 and *C. albicans* ATCC 10231, respectively; a GID of about 10 mm was present for all the other tested bacteria. In the case of *P. anisum*, the observed GID ranged from 10 ± 0.4 mm (E. coli ATCC 35218) to 13 ± 0.2 mm (*L. monocytogenes* ATCC 7644). The other EOs exhibited lower GID values (<10 mm) and, in some cases (such as *F. gummosa* or *J. communis*), were active only against some of the examined microorganisms (i.e., *S. aureus* ATCC 29213 and *P. fluorescens* DSM 4358). *C. carvii* EO showed no relevant activity against the panel of tested microorganisms (Gram-positive, Gram-negative bacteria, and mycetes), as well as *C. sativa* EO that were active only against *C. albicans* ATCC 10231 (GID 25 ± 1.5 mm). Based on these data, *C. maritimum*, *C. cyminum* and *P. anisum* EOs, showing the highest GID values, were selected to further formulate the respective MEs.

### 2.2. Chemical Composition of the Tested EOs

The chemical composition of the selected EOs is reported in Appendix A. As reported above, the most active EOs were those extracted from *C. cyminum*, *C. maritimum* and *P. anisum*. These EOs were characterized by different chemical profiles (Figure 2) that can justify the diverse rates of inhibition displayed on the different microbial strains. The *C. cyminum* EO was characterized by the oxygenated monoterpenes cumin aldehyde (30.2%) and γ-terpinen-7-al (17.3%), and the monoterpene hydrocarbons γ-terpinene (12.2%), *p*-cymene (10.2%), and β-pinene (9.8%). The *C. maritimum* EO was characterized by the monoterpenoids γ-terpinene (32.9%), carvacrol methyl ether (21.9%), and the phenylpropanoid dill apiole (17.5%). The EO composition of *P. anisum* was almost all dominated by the phenylpropanoid (E)-anethole (96.7%).

The chemical profiles detected in these samples were partly overlapping those reported in previous studies [26,27,28].

### 2.3. Formulation and Characterization of EO-MEs

All the MEs were prepared with an already tested mixture of excipients comprising ethanol, polysorbate (Tween 80), glycerol, and water [29]. Different ratios were evaluated before considering the final mixture composed of 20% *w*/*w* of ethanol, 20% *w*/*w* of polysorbate, 15% *w*/*w* of glycerol, 40% *w*/*w* of water, and 5% *w*/*w* of EOs. This mixture allowed the production of transparent and stable MEs with all the three considered EOs (*C. maritimum*, *C. cyminum* and *P. anisum* EOs) (Figure 3). The formulated MEs were analyzed through a DLS technique showing a bimodal size distribution as shown in Table 2. In all cases, the fraction of the populations with the smallest droplet size (ranging from 40 to 70 nm) was predominant. The presence of a second particle population with a larger droplet diameter (250–450 nm) in the intensity plot corresponded to a marginal fraction of the whole population. These two peaks could be related to the colloidal droplets which incorporated the EOs. The droplet diameters were not significantly affected after one month of storage at room temperature in the dark (data not shown).

### 2.4. Antimicrobial Activity of Free EOs Compared to the Formulated EO-MEs

The MIC and MBC values of the selected EOs in comparison to the related formulated EO-MEs are reported in Table 3. As a general trend, MEs performed better than the related EOs in all the representative microorganisms used in this study. As evidence, among the examined formulations, the *C. cyminum*-ME showed the highest antimicrobial activity, displaying a two-time lower (0.312%) MIC compared to the free EO (0.625%), and was even eight times lower against S. aureus ATCC 29213 (0.156 vs. 1.25%). A more remarkable antimicrobial effect was also noted in terms of MBC values that were eight times lower (from 0.312 to 0.625%) compared to the free EO (from 2.5 to 5%), and four times lower (0.312 vs. 1.25%) only in the cases of *P. fluorescens* DSM 4358. Similarly, the *P. anisum*-ME showed two times lower MIC values (1.25%) in the case of *E. coli* ATCC 35218, *L. monocytogenes* ATCC 7644 and *S. aureus* ATCC 29213 compared to the EO (2.5%), reaching an MIC four times lower (0.132%) for *P. fluorescens* DSM 4358 and *C. albicans* ATCC 10231 (free oil MIC 1.25%). Interestingly, the MBC values of *P. anisum*-ME were 2.5% for most of the examined microorganisms and 1.25% in the case of *P. fluorescens* DSM 4358, contrary to the free EO with MIC values always >5%.

With regards to the *C. maritimum*-ME, a two-time lower MIC (1.25%) was observed against *E. coli* ATCC 35218, while for *P. fluorescens* DSM 4358 and *C. albicans* ATCC 10231 a four-time lower MIC was obtained (1.25% and 0.312%) compared to the related EO (2.5 and 1.25%). On the contrary, no difference was observed for *L. monocytogenes* ATCC 7644 and *S. aureus* ATCC 29213 (1.25% for both ME and free EO). As shown in Table 3, an MBC value of 2.5% was observed for *E. coli* ATCC 35218, *L. monocytogenes* ATCC 7644, and *S. aureus* ATCC 29213 in comparison to the MBC values ≥5% of the free EO, reaching an MBC of 1.25% in the case of *P. fluorescens* DSM 4358 and *C. albicans* ATCC 10231 (a four-time lower MBC).

As shown, MBC/MIC ratio evidenced, in most cases, the bactericidal effect (MBC/MIC ≤ 4) of the examined MEs, with the exception of *P. anisum*-ME against *C. albicans* (MBC/MIC > 4).

## 3. Discussion

The advantages coming from the formulation of MEs such as an increased dispersion of hydrophobic molecules in aqueous environments have been successfully applied in various fields (e.g., foods, pharmaceuticals, cosmetics, agrochemicals, etc.). Indeed, the formulation of EO-MEs allows one to achieve concentrations of EO unattainable after simple dispersions. MEs are characterized by a nanometric size of the dispersed phase conferring peculiar features such as enhanced thermodynamic stability, increased solubility and bioavailability of active ingredients, dispersion uniformity, and increased penetration of actives into the target site [30]. As a consequence, the final effect of the EO-ME is a recognized potent antibacterial activity against both Gram-positive and Gram-negative bacteria [31,32].

Starting from these considerations, we aimed to better understand to what extent the formulations of EO-MEs can enhance the EOs’ antimicrobial activity, by performing a comparative analysis of three selected EOs and the related MEs against foodborne pathogens as well as spoilage and contaminant microorganisms. The selection of the most active EOs was based on the antimicrobial activity shown in AWDM by eleven different EOs tested at 10% (*v*/*v*). This assay allowed us to verify that not all the examined EOs showed the same activity and, indeed, some of them (such as *C. sativa* and *C. carvi*) had no relevant inhibitory effect on the growth of tested microorganisms. For this reason, only the EOs with a GID >10 mm against both bacteria and mycetes were used to formulate the MEs, specifically *C. maritimum*, *C. cyminum*, and *P. anisum* EOs. These EOs and some of their components have a well-demonstrated antimicrobial activity against several pathogens [33,34] as well as against different types of fungi and mycetes [35]. The EOs and their components specifically act on the external envelope of the cell as well as the cytoplasm. In terms of the mechanism of action, this includes a degradation of the cell wall, damage of the cytoplasmic membrane and associated proteins, an increase of permeability with consequent leakage of cell content, and cytoplasm coagulation [7,36]. For these reasons, the antimicrobial activity of the EOs cannot be attributed only to a single mechanism but can be considered the sum of a cascade of reactions involving the whole bacterial cell. 

In the second part of the presented study, the most promising EOs (i.e., from *C. maritimum*, *C. cyminum*, and *P. anisum*) were formulated as EO-MEs, using a mixture of excipients already used in similar formulations [15]. Excipient composition was modified to obtain transparent and stable MEs. Tween 80 was utilized as surfactant, since it is commonly employed in the formulation of MEs and presents some antimicrobial properties per se [37]. Glycerol and ethanol were added as biocompatible and biodegradable stabilizers. Among nanotechnological formulations, MEs are the most cost-effective to produce and the easiest to formulate and handle. The antimicrobial activity of each prepared ME and its related EO were determined and compared. Most of the data available in the literature refer to a single EO and its formulation [36,38,39,40,41], while few research works are focused on testing more EOs and their related formulations, contemporarily [30,42]. Therefore, we decided to study three different EOs to investigate the real applicability of the ME as a tool to increase their antimicrobial efficacy. The comparative analysis of the results, assessed by MIC and MBC determination, has evidenced that in most cases the MEs are more active than the corresponding EO. Indeed, an improvement in the antimicrobial performance was observed using the *C. maritimum*-ME, with a remarkable lowering of the MIC (between 0.312 and 1.25%) and MBC (1.25 and 2.5%) values compared to those of the EO (MIC values between 1.25 and 2.5% and MBC often >5%). This trend appeared even more marked in the case of the *C. cyminum*-EO, whose EO showed the lowest MIC values (between 0.625 and 1.25%). Interestingly, the *C. cyminum*-ME resulted in a further lowering of the MICs (between 0.156 and 0.312%) and an improvement of the microbicidal activity, with MBC values ranging between 0.625 and 0.312%, lower than those of the corresponding EO. In the case of the *P. anisum*-ME, a lowering of MIC values between 0.312 and 1.25% compared to those of the EO (between 1.25 and 2.5%) was observed, and notably, MBC values ranging from 1.25 to 2.5% were reached while MBC values always >5% were obtained with the corresponding EO. The comparison of the antimicrobial activity of EO-MEs showed similar activities against the Gram-positive and Gram-negative bacteria included in this study (similar lowering of the MIC and MBC values), indicating that the formulated MEs can be efficiently transported through the outer membrane porins of Gram-negative bacteria; therefore, the MEs size and the exposition of the hydrophilic groups of the emulsifying molecules allow and even potentiate an effective delivery of the EOs to the cell [31]. 

Interestingly, the obtained data stress the importance of the aqueous nanodispersions to overcome the limits of the direct use of EOs, in line with other authors [36,43]. Indeed, in the present work, the antibacterial activity of the *C. cyminum* EO is mostly attributable to the high level of cuminaldehyde (30.2%), a compound with well-known antimicrobial properties [44,45], while some other components, such as γ-terpinene, p-cymene, and β-pinene, representing a total of 49.5%, probably contribute to the remarkable antimicrobial activity of this EO, as also observed by other authors [46,47,48]. Similarly, (E)-anethole (96.7%) is responsible for the good antimicrobial activity of *P. anisum* EO as evidenced by Al Hafi et al. [49], while in the *C. maritimum* EO, there are many compounds (mostly monoterpenoids, carvacrol, and phenylpropanoids) that can be considered active against microorganisms [50]. From the comparison of the data, it can be clearly evidenced that the Eos’ pre-existent antimicrobial activity was further enhanced in the formulated MEs. The better performance of the examined MEs, even at lower concentrations compared to the related EOs, can be explained considering some peculiarities of the ME [30]. At first, the small size of the dispersed phase (ranging from 40 to 70 nm) may facilitate contact with the microbial surface structures. Moreover, the ability of MEs to spread the EOs into the growth medium probably increased the concentration of active ingredients at the interface [51]. Secondly, the increased surface area of the EOs’ droplets in the MEs causes a higher passive cellular absorption and an easier penetration of the active compounds into the target site. However, in some studies, no change in the antimicrobial activity of emulsified EOs was reported [52,53,54]. An explanation could be the possible incorporation of a carrier oil phase (such as corn oil and soybean oil) along with the EOs in the antimicrobial delivery systems. The carrier oils have no antimicrobial activity per se, but rather act as solvents for the EOs, thus reducing the antibacterial activity of the ME, and allowing a lower amount of EO to interact with the target bacteria [52,53]. 

## 4. Materials and Methods

### 4.1. Plant Material and EOs Preparation

EOs were obtained from eleven plant sources including 10 species (*Cannabis sativa*, *Carum carvi*, *Crithmum maritimum*, *Cuminum cyminum*, *Cupressus arizonica*, *Ferula assa-foetida*, *Ferula gummosa*, *Juniperus communis* and *Pimpinella anisum*) and two hybrids (x *Cupressocyparis leylandii* and *Juniperus* x *pfitzeriana*). They were obtained from spontaneous and cultivated accessions; only in the case of *C. carvi* was a commercial sample used. Complete information on these samples is reported in Table 4. Plant materials, except for *C. carvi* that was a ready-to-use sample, were subjected to hydrodistillation using a Clevenger-type apparatus. The heating system was a Falc MA (Falc, Instruments, Treviglio, Italy) and the amount of distilled water ranged from 5 to 7 L, depending on the sample amount. The time of distillation was 3 h (until no more oil condensed). Once decanted (20 min), the EOs were separated from the aqueous layer, then treated with anhydrous sodium sulphate to remove water drops, filtered, and stored in dark vials at −20 °C before use. EO yields were determined on a dry weight basis, as reported in Table 4.

Stock solutions (10% *v*/*v*) of each EO were prepared in polyethylene glycol (PEG) and ethanol (50:50, *v*/*v*) (Sigma, Milan, Italy) and kept at −20 °C in the dark, until use.

### 4.2. Chemical Analysis of EOs Composition

The chemical composition of each EO was obtained on a gas chromatograph 6890N equipped with a 5973N single quadrupole mass spectrometer and an autosampler 7863 from Agilent Technologies, Santa Clara, CA, USA. The temperature of the injector and detector was set to 280 °C. Separation of volatiles was achieved on an HP-5MS capillary column (30 m × 0.25 mm, 0.1 µm) coated with 5% phenylmethylpolysiloxane and heated in an oven with the following ramp: 60 °C (5 min) to 220 °C at 4 °C min^−1^ and then 11 °C min^−1^ to 280 °C (15 min). The mobile phase was He (99.999%) flowing at 1 mL min^−1^. All samples were diluted 1:100 in analytical-grade hexane (Sigma-Aldrich, Milan, Italy) and 1 µL of the solution was injected using the split mode (1:50). The electron energy of the detector was 70 eV scanning in the range 29–400 *m*/*z*. The peak assignment was based, whenever possible, on the comparison of analytical standards available in the authors’ laboratory. In addition, the confirmation was based on the calculation of the retention index according to the Van den Dool and Kratz [55] formula and comparison with those reported in Adams [56] and NIST 17 [57]. Similarly, the mass spectrum of each peak was compared with those stored in the aforementioned libraries plus the FFNSC3 [58]. Semiquantification of components was based on the relative peak areas without using correction factors.

### 4.3. Microbial Strains, Culture Conditions, and Inoculum Preparation

Four reference strains belonging to the American Type Culture Collection (ATCC, Rockville, MD, USA) were used in this study: *E. coli* ATCC 35218, *L. monocytogenes* ATCC 7644, *S. aureus* ATCC 29213, and *C. albicans* ATCC 10231. *P. fluorescens* DSM 4358 was obtained from the German Collection of Microorganisms and Cell Culture (DSMZ, Braunschweig, Germany). Bacterial strains were routinely grown in tryptic soy agar (TSA, VWR, Milan, Italy) at 37 °C for 24 h, with the exception of *P. fluorescens* DSM 4358, for which the growth temperature was set to 30 °C. *C. albicans* ATCC 10231 was grown on Sabouraud dextrose agar (SDA, VWR) at 37 °C for 48 h. All strains were stored at −80 °C in Nutrient Broth no. 2 (VWR) with 15% (*v*/*v*) of glycerol.

To prepare the inocula, all strains were grown in 10 mL of tryptic soy broth (TSB, VWR) overnight at 37 °C or 30 °C (*P. fluorescens* DSM 4358). Then, the number of cells necessary for each experiment was determined spectroscopically (OD610 nm ca. 0.13–0.15) and adjusted to 5 × 10^6^ cfu/mL (colony forming unit per milliliter). These bacterial suspensions were used in all the following experiments. 

### 4.4. Screening of EOs Antimicrobial Activity by Agar Well Diffusion Method

Initially, the susceptibility of each foodborne pathogen and spoilage microorganism to EOs was determined using the agar well diffusion method (AWDM) according to Campana et al. [17], with some modifications. Briefly, each bacterial cell suspension, prepared as described above, was diluted in the 1:10 ratio in Mueller–Hinton broth II (MHB, VWR), gently seeded on the surface of 25 mL Mueller–Hinton agar (MHA) (VWR) plates and then kept at room temperature under flow cabinet for at least 10 min. At this point, wells of 6 mm in diameter were made on the agar with sterile stainless-steel cylinders, and 40 μL of each EO (10%, *v*/*v*) was dropped into the holes; in each plate, two holes were filled with 40 μL of polyethylene glycol (PEG)—ethanol (50:50, *v*/*v*) or 40 μL of Mueller–Hinton broth II (MHB, VWR), respectively, as controls. Gentamicin (10 μg) and ciprofloxacin (5 μg) disks (Oxoid, Milan, Italy) were included as positive controls for bacteria, while fluconazole disks (50 and 25 μg/mL) were used for *C. albicans*. All plates were incubated for 24 h at 37 °C, then the growth inhibition diameter (GID) around each hole was measured (in mm). In this study, we consider a GID value greater than 10 mm indicative of antibacterial activity of the examined EOs. Independent experiments were performed in duplicate. As mentioned, for *P. fluorescens* DSM 4358 the growth temperature was 30 °C.

### 4.5. Microemulsion Formulation and Characterization

The EOs-based MEs were produced by controlled addition of the EO phase into the water phase under magnetic stirring. The microemulsions were composed as follow: 5% *w*/*w* EO, 15% *w*/*w* glycerol (Galeno srl, Comeana, Italy), 20% *w*/*w* pure ethanol (Merck, Italy), 20% *w*/*w* polysorbate 80 (Tween^®^ 80, Sigma Aldrich, Milan, Italy) and 40% *w*/*w* ddH_2_O.

To obtain a stable microemulsion, firstly, the EO was mixed with ethanol and loaded in a syringe to be injected into the water phase using a syringe pump (Aladdin syringe pump, WPI, Berlin, Germany) at a controlled flow rate of 0.05 m min^−1^. The water phase was composed of glycerol, polysorbate 80, and ddH_2_O constantly stirred at 600 rpm. The formulated MEs obtained with this method were transparent. To determine the hydrodynamic diameter (size) of the oily internal phase of each ME, dynamic light scattering (DLS) analysis using a Malvern Zetasizer Nano S (Malvern instrument, Malvern, UK) was performed. DLS technique allows the determination of the size distribution profile of small droplets present in emulsions by evaluating the scattering of a laser light at an angle of 173°. One milliliter of each sample was loaded into disposable cuvettes and the samples were analyzed at 25 °C. Before each analysis, the samples were balanced for 180 s at the selected temperature. The stability of the formulations was evaluated with the same technique after one month of storage at room temperature. In addition, controls without EOs were prepared.

### 4.6. Comparison of the Antimicrobial Activity of EOs and Formulated EO-MEs

For each selected EO and the related EO-MEs, MICs as well as MBCs were determined by the broth microdilution method against all the examined microorganisms [59]. Each overnight microbial culture, obtained as described above, was diluted in MHB II (VWR) to obtain ca. 5 × 10^5^ cfu/mL; then 50 μL was added to a 96-well microtiter plate (Cellstar, Greiner bio-one) in the presence of 50 μL of each EO (serially diluted from 5 to 0.156%, *v*/*v*) or 50 μL of each EO-MEs (serially diluted from 2.5 to 0.156%, *v*/*v*). Positive controls (i.e., bacteria in MHB II) and negative control (i.e., uninoculated MHB II) were also included. The MIC was defined as the lowest concentration of tested EOs and related MEs inhibiting visible growth after 24 h of incubation [59]. To determine the MBC values, 10 μL from the invisible growth was inoculated in triplicate on TSA and incubated at 37 °C for 24 h; MBC was defined as the lowest concentration of EO that inhibited growth rebound on TSA. All experiments were performed in duplicate.

## 5. Conclusions

In conclusion, in spite of the well-demonstrated antimicrobial potential of EOs in vitro, their use has been limited because high concentrations are required to achieve sufficient antimicrobial activity [44]. In this study, EOs-MEs were developed and then evaluated for antimicrobial effects against food-borne pathogens and spoilage microorganisms. The obtained results indicated the potential of MEs to enhance the natural antimicrobial activity of the studied EOs, with the important advantages of improving their solubility and lowering the antimicrobial concentration necessary for a microbicide action. This study represents the first step toward the engineering of a novel and applicable antimicrobial system based on EO-MEs, facilitating a closer interaction with microorganisms and EO-active compounds. 

## Figures and Tables

**Figure 1 antibiotics-11-00447-f001:**
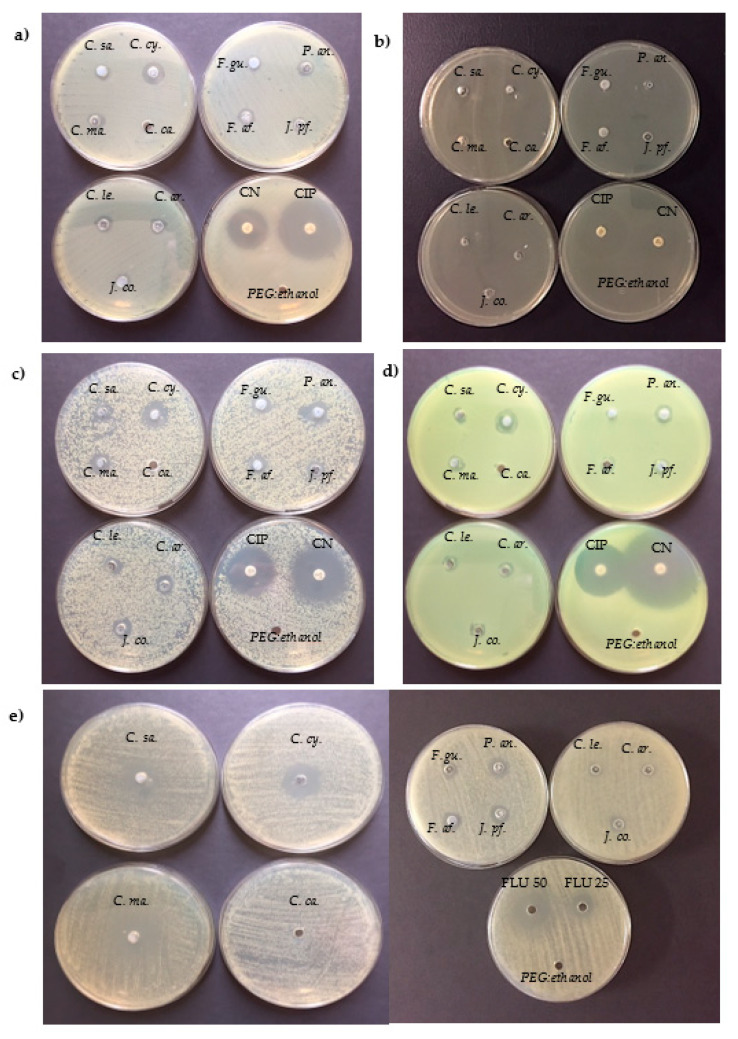
Representative images of growth inhibition diameter (GID; mm) produced by the eleven essential oils (EOs, 10% *v*/*v*) against *E. coli* ATCC 35218 (**a**), *L. monocytogenes* ATCC 7644 (**b**), *S. aureus* ATCC 29213 (**c**), *P. fluorescens* DSM 4358 (**d**), and *C. albicans* ATCC 10231 (**e**) determined by AWDM. GID of antibiotics (gentamicin 10 µg and ciprofloxacin 5 µg) or antifungal (fluconazole 50 µg and 25 µg) as well as PEG:ethanol 50% (used as vehicle) were also presented. *C.sa, Cannabis sativa; C.ca., Carum carvi; C.ma., Crithmum maritimum; C.cy., Cuminum cyminum; C.le; x Cupressocyparys leylandii; C.ar., Cupressus arizonica; F.a.f., Ferula assa-foetida; F.gu., Ferula gummosa; J.co., Juniperus communis; J.pf., Juniperus x pfitzeriana; P.an., Pimpinella anisum*.

**Figure 2 antibiotics-11-00447-f002:**
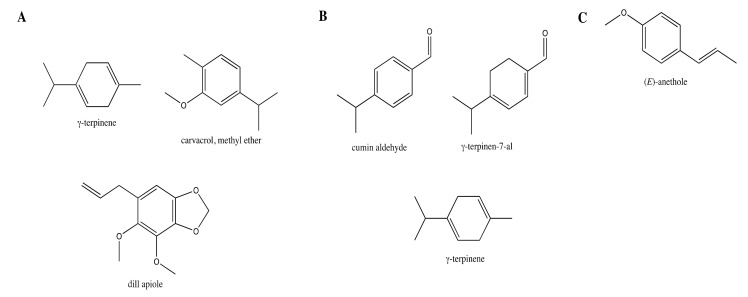
Major volatile components of *C. maritimum* oil (**A**), *C. cyminum oil* (**B**), and *P. anisum* oil (**C**) identified by GC/MS analysis. Minor compounds (<10%) are not shown.

**Figure 3 antibiotics-11-00447-f003:**
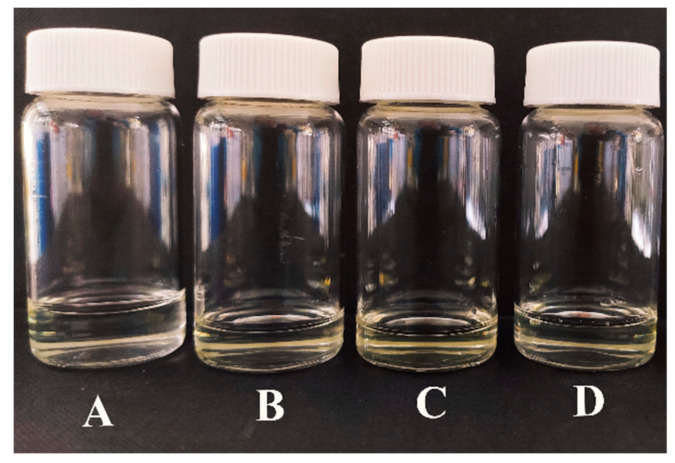
Microemulsions formulated in this study: blank control solution without EOs (**A**); *C. marittimum*-ME (**B**); *C. cyminum*-ME (**C**); *P. anisum*-ME (**D**). The formulated EO-MEs (5% *w*/*w*) are limpid and transparent and, in some cases, with a slight yellowish color.

**Table 1 antibiotics-11-00447-t001:** Antimicrobial activity of the examined eleven essential oils (EOs) (10% *v*/*v*) against foodborne pathogens and spoilage microorganisms determined by AWDM. Antibiotics (gentamicin and ciprofloxacin), or antifungal (fluconazole) as well as PEG:Ethanol 50% were used as controls. Data represent the growth inhibition diameter (GID; mm; mean ± sd) of independent experiments performed in duplicate.

EOs (10% *v*/*v*)	*E. coli* ATCC 35218	*L. monocytogenes* ATCC 7644	*S. aureus* ATCC 29213	*P. fluorescens* DSM 4358	*C. albicans* ATCC 10231
*C. sativa*	0	10 ± 0.5	0	0	25 ± 1.2
*C. carvi*	0	0	8 ± 0.4	8 ± 0.2	0
*C. maritimum*	10 ± 0.5	10 ± 0.6	13 ± 0.4	10 ± 1.2	25 ± 1.5
*C. cyminum*	12 ± 1.0	11 ± 2.1	15 ± 1.4	12 ± 1.4	25 ± 1.5
*C. leylandii*	10 ± 0.1	10 ± 0.2	10 ± 0.4	9 ± 0.2	0
*C. arizonica*	9 ± 1.4	9 ± 0.1	10 ± 0.4	8 ± 0.2	8 ± 0.4
*F. assa-foetida*	10 ± 0.6	8 ± 0.2	11 ± 0.4	0	9 ± 0.5
*F. gummosa*	0	8 ± 0.1	11 ± 0.4	10 ± 0.4	10 ± 0.3
*J. communis*	0	9 ± 0.4	10 ± 0.6	7 ± 0.5	8 ± 0.6
*J. pfitzeriana*	0	8 ± 0.5	9 ± 0.1	10 ± 0.2	7 ± 0.2
*P. anisum*	10 ± 0.4	13 ± 0.2	12 ± 1.1	10 ± 0.5	11 ± 1.4
PEG:ethanol 50%	0	0	0	0	0
Ciprofloxacin 5 µg	>30	27 ± 2.1	>30	>30	-
Gentamicin 10 µg	24 ± 1.7	>30	26 ± 1.8	26 ± 0.9	-
Fluconazole 25 µg	-	-	-	-	12 ± 1.5
Fluconazole 50 µg	-	-	-	-	25 ± 2.2

**Table 2 antibiotics-11-00447-t002:** Z-average sizes (nm) of the two populations present in each formulation measured with the DLS technique.

Sample	Size Peak 1 (nm)	Size Peak 2 (nm)
*C. maritimum*-ME	43	450
*C. cyminum*-ME	49	246
*P. anisum*-ME	47	315

**Table 3 antibiotics-11-00447-t003:** Comparative MIC and MBC values (%, *v*/*v*) of the EOs and related MEs against foodborne pathogens and spoilage microorganisms. MIC and MBC values of EO-MEs lower than those of the related EOs are highlighted in grey. Fold-reduction of MIC and MBC values of each EO in the formulated ME as well as the related MBC/MIC ratio are indicated.

	*E. coli* ATCC 35218			*L. monocytogenes* ATCC 7644			*S. aureus* ATCC 29213			*P. fluorescens* DSM 4358			*C. albicans* ATCC 10231		
	MIC	MBC	MBC/MIC	MIC	MBC	MBC/MIC	MIC	MBC	MBC/MIC	MIC	MBC	MBC/MIC	MIC	MBC	MBC/MIC
*C. maritimum*:															
EO	2.5	>5	nd	1.25	>5	nd	1.25	>5	nd	1.25	5	4	1.25	5	4
ME	1.25	2.5	2	1.25	2.5	2	1.25	2.5	2	0.312	1.25	4	0.312	1.25	4
Fold-reduction	2	nd		-	nd		-	nd		4	4		4	4	
*C. cyminum*:															
EO	0.625	2.5	4	0.625	2.5	4	1.25	2.5	2	0.625	1.25	2	0.625	5	8
ME	0.312	0.312	1	0.312	0.312	1	0.156	0.312	2	0.312	0.312	1	0.312	0.625	2
Fold-reduction	2	8		2	8		8	8		2	4		2	8	
*P. anisum*:															
EO	2.5	>5	nd	2.5	>5	nd	2.5	>5	nd	1.25	>5	nd	1.25	>5	nd
ME	1.25	2.5	2	1.25	2.5	2	1.25	2.5	2	0.312	1.25	4	0.312	2.5	8
Fold-reduction	2	nd		2	nd		2	nd		4	nd		4	nd	

nd: not determined because the MBC values of the EO was >5%*;* MBC/MIC ≤ 4: bactericidal effect; MBC/MIC > 4: bacteriostatic effect.

**Table 4 antibiotics-11-00447-t004:** Plant sources of the essential oils investigated in this work.

Plant Species	Abbreviation	Family	Part Used	Origin and Status	Collection Site and Year	Oil Yield (%, *w*/*w*)
*Cannabis sativa* L. cv CS	*C. sativa*	Cannabaceae	Female inflorescences	Italy, cultivated	Fiuminata (Italy), 2018	0.3
*Carum carvi* L.	*C. carvi*	Apiaceae	Fruits (schizocarps)	Pakistan, commercial sample (Hemani International KEPZ)	Kafarkila (Lebanon), 2018	nr
*Crithmum maritimum* L.	*C. maritimum*	Apiaceae	Flowering aerial parts	France, wild	Le Conquet (Bretagne), 2018	0.8
*Cuminum cyminum* L.	*C. cyminum*	Apiaceae	Fruits (schizocarps)	Syria, cultivated	Syria, 2018	3.2
x *Cupressocyparis leylandii* A.B.Jacks. & Dallim.	*C. leylandii*	Cupressaceae	Green twigs	Italy, cultivated	Pratola Peligna (Italy), 2018	1.0
*Cupressus arizonica* Greene	*C. arizonica*	Cupressaceae	Green twigs	Italy, wild	Pratola Peligna (Italy), 2016	0.6
*Ferula assa-foetida* L.	*F. assa-foetida*	Apiaceae	Oleo-gum-resin	Iran, wild	Kohsorkh, 2019	8.9
*Ferula gummosa* Boiss.	*F. gummosa*	Apiaceae	Oleo-gum-resin	Iran, wild	Kohsorkh, 2019	13.7
*Juniperus communis* L.	*J. communis*	Cupressaceae	Green twigs	Italy, wild	Sulmona (Italy), 2018	0.6
*Juniperus* x *pfitzeriana* (Späth) P.A.Schmidt	*J. pfitzeriana*	Cupressaceae	Green twigs	Italy, cultivated	Sulmona (Italy), 2018	2.0
*Pimpinella anisum* L.	*P. anisum*	Apiaceae	Fruits (schizocarps)	Italy, cultivated	Castignano (Italy), 2017	2.4

nr: not reported.

## Data Availability

Data is contained within the article.

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
