# Peer review of "Comparative Analysis of the Antimicrobial Activity of Essential Oils and Their Formulated Microemulsions against Foodborne Pathogens and Spoilage Bacteria"

_antibiotics, 2022, doi:10.3390/antibiotics11040447_

Round 1

Reviewer 1 Report

The manuscript is well written and sounds interesting. It would be a better manuscript with just a few corrections. 1. The first word should not be abbreviated. ex) MIC, MBC 2. The authors need to explain how they determined the composition of microemulsion because results may vary depending on microemulsion configuration. It would be even better if there are comparative experimental results for this.

Author Response

The manuscript ID: antibiotics-1650172 was revised as suggested by the reviewers. New references were added. The authors thanks for all the suggestions and tried to follow them to improve the quality of the manuscript.

Reviewer n. 1

The manuscript is well written and sounds interesting. It would be a better manuscript with just a few corrections.

  1. The first word should not be abbreviated. ex) MIC, MBC
  2. The authors need to explain how they determined the composition of microemulsion because results may vary depending on microemulsion configuration. It would be even better if there are comparative experimental results for this.

Our responses: Thanks for the suggestions. We have corrected the term MIC and MBC adding the full names. As regards the ME composition, the formulations have been previously optimized as reported with reference 23 and the composition is 20% w/w of ethanol, 20% w/w of polysorbate, 15% w/w of glycerol, 40% w/w of water, and 5% w/w of EOs. The formulation was done with a controlled injection of a mixture containing the EOs mixed with ethanol into water containing polysorbate and glycerol as reported in section 4.5. All the EO microemulsions have been compared to a control made with the same excipients without the EO. MEs were characterized using DLS and the stability was assessed after one month of storage at room temperature in the dark.

Reviewer 2 Report

The work is interesting and well written. After minor revision, it could be considered for publication in the Journal. Please, find attached my comments.

Author Response

Responses to Reviewers

The manuscript ID: antibiotics-1650172 was revised as suggested by the reviewers. New references were added. The authors thanks for all the suggestions and tried to follow them to improve the quality of the manuscript.

Reviewer n. 2

We have checked the manuscript and, following the indication given by the reviewer, we have made all the corrections required.

Specifically, the term v/v or w/w now are now written in italic, and the bacterial species names are abbreviated in brackets. The full names of MIC and MBC were given in the abstract.

Information on plants were given in the Abstract as well as in the Introduction with the related references (also added in the Reference List).

The Supplementary Material contained an error (the images of agar plates) that was corrected. We thank for the information.

We are aware that in Figure 1 is not well evident the growth of Listeria monocytogenes, but is only because the colonies appeared small and translucent, visible only against the light and really difficult to see when directly exposed to light (as happened for the photos). In any case, the data reported in Table 1 are the results of the experiments performed in duplicate and each time the growth of Listeria was similar. The assay was carried on using the same agar media for all the examined microorganisms and we have not changed the protocol selecting a probably more suitable media (as TSAYE) for Listeria.

In Table 1 the sd for antibiotics were added.

In Table 3 the MBC/MIC ratio was added and briefly commented in the results.